# STYLE MEMORY: MAKING A CLASSIFIER NETWORK GENERATIVE

## ABSTRACT

Deep networks have shown great performance in classification tasks. However, the parameters learned by the classifier networks usually discard stylistic information of the input, in favour of information strictly relevant to classification. We introduce a network that has the capacity to do both classification and reconstruction by adding a "style memory" to the output layer of the network. We also show how to train such a neural network as a deep multi-layer autoencoder, jointly minimizing both classification and reconstruction losses. The generative capacity of our network demonstrates that the combination of style-memory neurons with the classifier neurons yield good reconstructions of the inputs when the classification is correct. We further investigate the nature of the style memory, and how it relates to composing digits and letters.

## 1 INTRODUCTION

Deep neural networks now rival human performance in many complex classification tasks, such as image recognition. However, these classification networks are different from human brains in some basic ways. First of all, the mammalian cortex has many feed-back connections that project in the direction opposite the sensory stream (Bullier et al., 1988). Moreover, these feed-back connections are implicated in the processing of sensory input, and seem to enable improved object/background contrast (Poort et al., 2012), and imagination (Reddy et al., 2011). Feed-back connections are also hypothesized to be involved in generating predictions in the service of perceptual decision making (Summerfield & De Lange, 2014).

Humans (and presumably other mammals) are also less susceptible to being fooled by ambiguous or adversarial inputs. Deep neural networks have been shown to be vulnerable to adversarial examples (Szegedy et al., 2014; Goodfellow et al., 2015). Slight modifications to an input can cause the neural network to misclassify it, sometimes with great confidence! Humans do not get fooled as easily, leading us to wonder if the feed-back, generative nature of real mammalian brains contributes to accurate classification.

In pursuit of that research, we wish to augment classification networks so that they are capable of both recognition (in the feed-forward direction) and reconstruction (in the feed-back direction). We want to build networks that are both classifiers and generative.

The nature of a classifier network is that it throws away most of the information, keeping only what is necessary to make accurate classifications. Simply adding feed-back connections to the network will not be enough to generate specific examples of the input – only a generic class archetype. But what if we combine the features of a classifier network and an autoencoder network by adding a "style memory" to the top layer of the network? The top layer would then consist of a classification component as well as a collection of neurons that are not constrained by any target classes.

We hypothesized that adding a style memory to the top layer of a deep autoencoder would give us the best of both worlds, allowing the classification neurons to contribute the class of the input, while the style memory would record additional information about the encoded input – presumably information not encoded by the classification neurons. The objective of our network is to minimize both classification and reconstruction losses so that the network can perform both classification and reconstruction effectively. As a proof of concept, we report on a number of experiments with MNIST and EMNIST that investigate the properties of this style memory.

## 2 RELATED WORK

Others have developed neural architectures that encode both the class and style of digits to enable reconstruction. Luo et al. (2017) recently introduced a method called bidirectional backpropagation. Their network is generative because it has feed-back connections that project down from the top (soft-max) layer. A digit class can be chosen at the top layer, and the feed-back connections render a digit of the desired class in the bottom layer (as an image). However, the network always renders the same, generic sample of the class, and does not reconstruct specific samples from the data.

Networks that have the capacity to generate images have been shown to learn meaningful features. Previous work (Hinton, 2007) showed that in order to recognize images, the network needs to first learn to generate images. Salakhutdinov & Hinton (2009) showed that a network consisting of stacked Restricted Boltzmann Machines (RBMs) learns good generative models, effective for pre-training a classifier network. RBMs are stochastic in nature, so while they can generate different inputs, they are not used to generate a specific sample of input. Bengio et al. (2006) also demonstrated that autoencoders pre-trained in a greedy manner also lead to better classifier networks. Both (Hinton et al., 2006) and (Bengio et al., 2006) use tied weights, where the feed-back weight matrices are simply the transpose of the feed-forward weights; this solution is not biologically feasible. These findings have inspired other successful models such as stacked denoising autoencoders (Vincent et al., 2010), which learn to reconstruct the original input image given a noise-corrupted input image.

Lastly, Salakhutdinov & Hinton (2007) also showed another method to map an input to a lower dimensional space that minimizes within-class distance of the input. They first pre-trained a network as RBMs, and then "unrolled" the network to form a deep autoencoder. The network was then fine-tuned by performing nonlinear neighbourhood component analysis (NCA) between the low-dimensional representations of inputs that have the same class. They were able to separate the class-relevant and class-irrelevant parts by using only 60% of the lower-dimensional code units when performing nonlinear NCA, but all the codes were used to perform reconstruction. As a result, their network was able to minimize within-class distance in the lower-dimensional space while maintaining good reconstruction. Inference was then performed by using K-nearest neighbour in that lower-dimensional space. Our method is similar, but our top layer includes an explicit classification vector alongside the class-agnostic style memory.

## 3 METHOD

### 3.1 MODEL DESCRIPTION

Our bidirectional network consists of an input layer, convolutional layers, fully connected layers, and an output layer. However, the output layer is augmented; in addition to classifier neurons (denoted by $y$ in Fig. 1), it also includes style-memory neurons (denoted $m$ in Fig. 1). A standard classifier network maps $x \in X \rightarrow y \in Y$, where the dimension of $Y$ is usually much smaller than the dimension of $X$. The feed-forward connections of our augmented network map $x \in X \rightarrow (y, m) \in Y \times M$. The output $y$ is the classification vector (softmax). The output $m$ is the style memory, meant to encode information about the particular form of an input. For the example of MNIST, the classification vector might represent that the digit is a '2', while the style memory records that the '2' was written on a slant, and with a loop in the bottom-left corner.

A classifier network can be trained as a deep autoencoder network. However, the decoder will only be able to generate a single, generic element of a given class. By adding a style memory in the output layer, the network will be able to learn to generate a variety of different renderings of a particular class.

### 3.2 TRAINING

We trained the network following a standard training procedure for deep autoencoders, depicted in Fig. 2. For the input layer, we follow the work from Vincent et al. (2010) by injecting small additive Gaussian noise to the input.

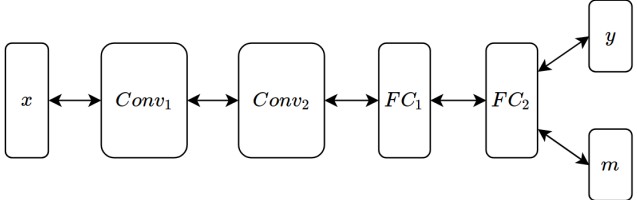

Figure 1: Our bidirectional network with a style memory in the output layer. Here, $x$ denotes the input ($x \in X$), while $Conv_i$ and $FC_i$ denote convolutional layer and fully connected layer $i$, respectively. Lastly, $y$ denotes output label ($y \in Y$), and $m$ denotes the style memory ($m \in M$).

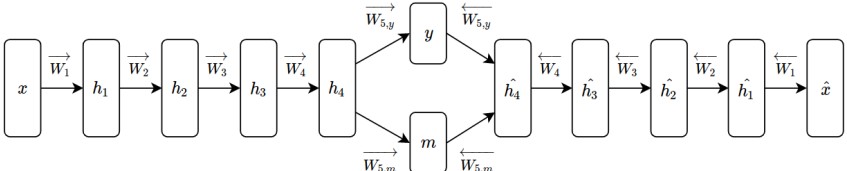

Figure 2: The "unrolled" network. Learning consists of training the network as a deep autoencoder, where $h_i$ denotes the hidden layer representation of layer $i$.

The objective for our network's top layer is to jointly minimize two loss functions. The first loss function is the classifier loss $L_y$, which is a categorical cross-entropy loss function,

$$L_y(y_t, y) = -\sum_x y_t \log(y) \,, \tag{1}$$

where $y_t$ is the target label, and $y$ is the predicted label. The second loss function is the reconstruction loss between the input and its reconstruction. This reconstruction loss, denoted $L_r$, is the Euclidean distance between the input to the top layer, and the reconstruction of that input,

$$L_r(\hat{x}, x) = \|\hat{x} - x\|_2 \,, \tag{2}$$

where $\hat{x}$ is the reconstruction of the input $x$, as shown in Fig. 2.

Our goal is to find connection weights, $W^*$, that minimize the combination of both loss functions in the last layer,

$$W^* = \arg\min_W \sum_{x \in X} L_y(y_t, y) + \alpha(L_r(\hat{x}, x)) \,, \tag{3}$$

where $W$ represents the parameters of the network, and $\alpha$ adjusts the weight of $L_r$.

## 4 Experiments

We performed all experiments in this paper using digits from MNIST and letters from Extended MNIST (EMNIST) (Cohen et al., 2017) datasets, with an input dimensionality of $28 \times 28$ pixels. The networks used for the experiments have two convolutional layers and two fully connected layers. The first and second convolutional layers are made of 32 and 64 filters, respectively. The receptive fields of both convolutional layers are $5 \times 5$ with a stride of 2, using ReLU activation functions. The fully connected layers $FC_1$ and $FC_2$ have 256 and 128 ReLU neurons, respectively.

The style memory consists of 16 logistic neurons, and the classifier vector contains either 10 or 26 softmax neurons, for MNIST or EMNIST, respectively. The reconstruction loss weight ($\alpha$) was set to 0.05, and the optimization method used to train the network was Adam (Kingma & Ba, 2014) with a learning rate $\eta$ of 0.00001 for 250 epochs. The network achieved 98.48% and 91.27% classification accuracy on the MNIST and EMNIST test sets, respectively.

### 4.1 Reconstruction Using Style Memory

The reconstructions produced by our network show that the network has the capacity to reconstruct a specific sample, rather than just a generic example from a specific class. Figures 3 and 4 show

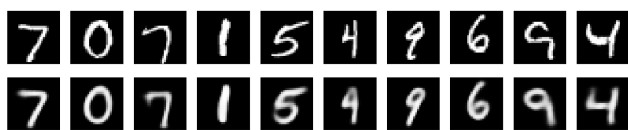

Figure 3: Reconstruction of MNIST digits using the network's predictions and style memories. The top row shows the original images from the MNIST test set, and the bottom row shows the corresponding reconstructions produced by the network.

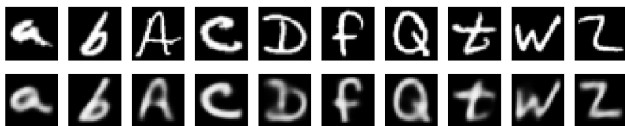

Figure 4: Reconstruction of EMNIST letters using the network's predictions and style memories.

examples of digit and letter reconstructions. Notice how the network has the ability to reconstruct different styles of a class, like the two different '4's, two different '9's, and two different 'A's. For each sample, the reconstruction mimics the style of the original character. Note that the digits and letters in both figures were correctly classified by the network.

## 4.2 RECONSTRUCTION OF MISCLASSIFIED SAMPLES

How do the softmax classification nodes and the style memory interact when a digit or letter is misclassified? The first column in Fig. 5 shows an example where the digit '3' was misclassified as a '5' with 71% confidence. The resulting reconstruction in the middle row looks more like a '5' (although there is a hint of a '3'). However, correcting the softmax neurons to the one-hot ground truth label for '3' changed the reconstruction to look more like a '3', as shown in the bottom row of Fig. 5. Similar results were observed when we used letters from the EMNIST dataset, as shown in Fig. 6.

We believe that the generative abilities of these classifier networks enable it to identify misclassified inputs. If the reconstruction does not closely match the input, then it is likely that the input was misclassified. This idea forms the crux of how these networks might defend against being fooled by adversarial or ambiguous inputs.

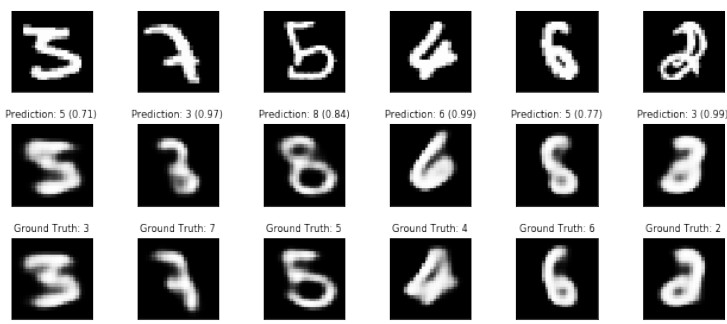

Figure 5: Comparison of MNIST digit reconstruction using the prediction from the network versus ground truth label. The top row shows the original images from the MNIST test set that the network misclassified. The middle row shows the reconstruction of the images, along with the incorrect class and confidence score. The bottom row shows the reconstructions using the corrected one-hot labels.

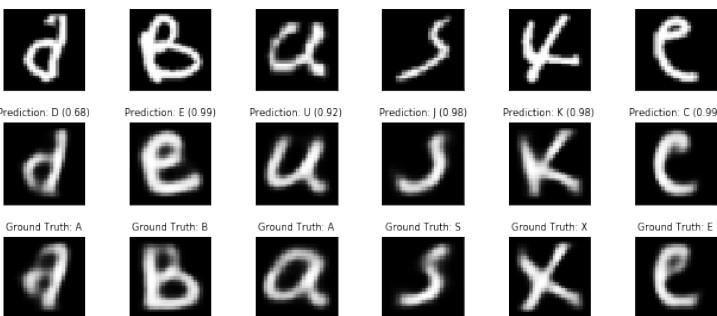

Figure 6: Comparison of EMNIST letter reconstruction using the prediction from the network versus ground truth label.

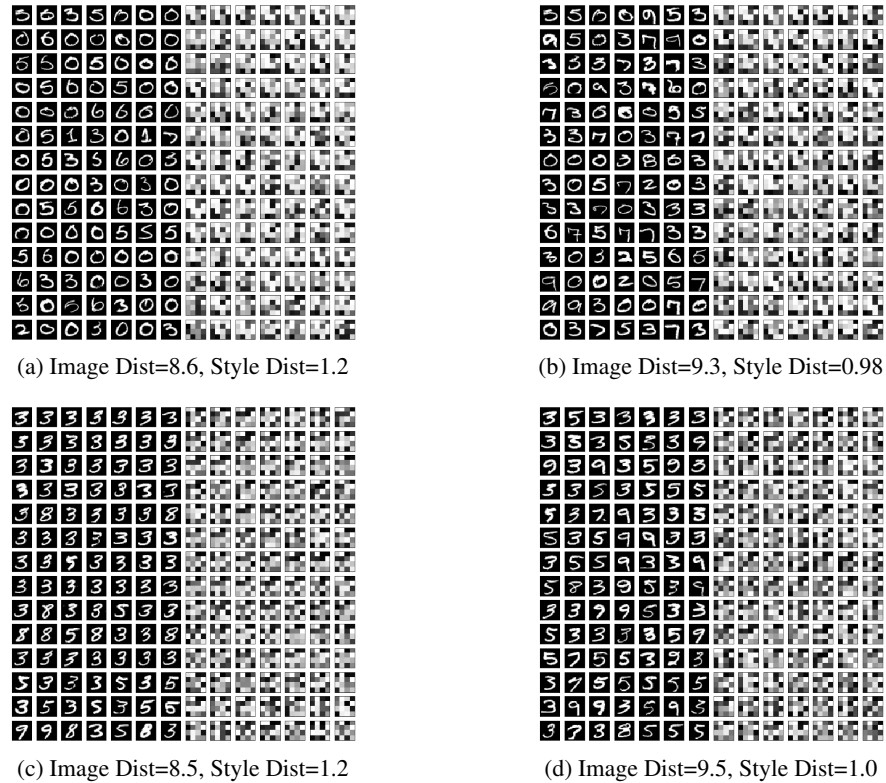

(a) Image Dist=8.6, Style Dist=1.2

(b) Image Dist=9.3, Style Dist=0.98

(c) Image Dist=8.5, Style Dist=1.2

(d) Image Dist=9.5, Style Dist=1.0

Figure 7: Nearest neighbours in image space and style-memory space. (a) and (c) show the 97 digit images closest to the image in the top-left, as well as their corresponding style-memories. (b) and (d) show the 97 style memories closest to the style memory in the top-left, as well as their corresponding digit images. The order of elements (across rows, then down) indicate increasing Euclidean distance. The subfigure captions give the average distance from the top-left element, both in image space, and style-memory space.

## 4.3 STYLE MEMORY REPRESENTATION

To better understand what was being encoded in the style memory, we generated digits that were close together in the style memory space (16-dimensional) and compared them with digits that are close together in the image space (784-dimensional). The distance, in either space, was calculated using the Euclidean norm.

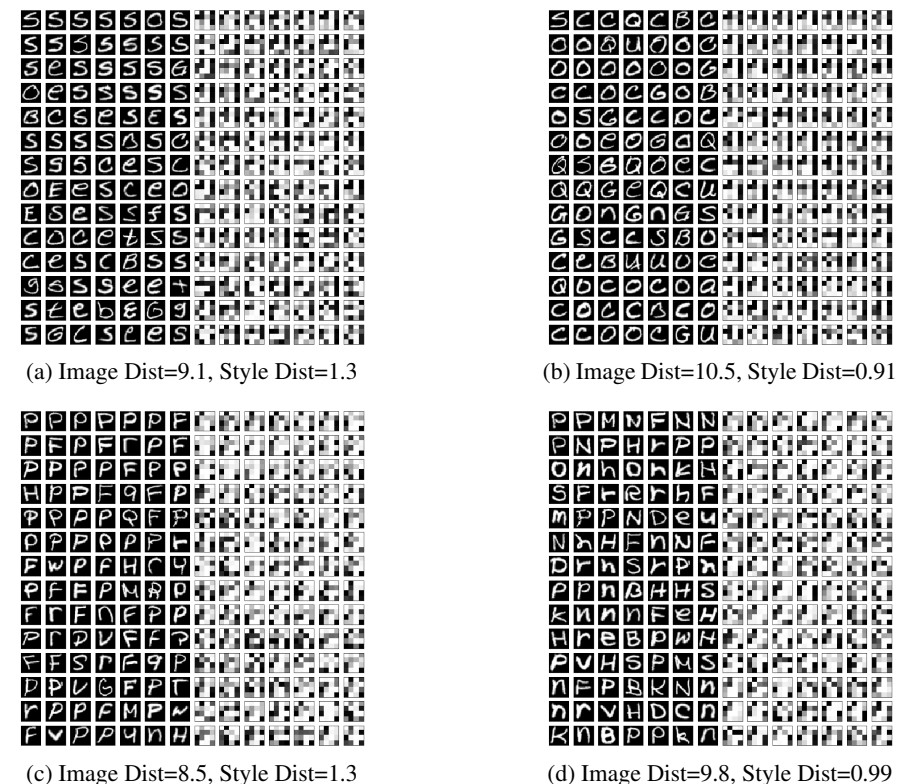

(a) Image Dist=9.1, Style Dist=1.3        (b) Image Dist=10.5, Style Dist=0.91

(c) Image Dist=8.5, Style Dist=1.3        (d) Image Dist=9.8, Style Dist=0.99

Figure 8: Nearest neighbours in image space and style-memory space of EMNIST dataset.

From Fig. 7 and Fig. 8, we can observe that proximity in the style-memory space has different semantic meaning than proximity in the image space. Figure 7a, showing the 97 images that are closest to the '5' image in the top-left corner, displays many digits that share common pixels. However, Fig. 7b, which shows the 97 digits with the closest style memories, displays digits that come from various different classes. Similarly, Fig. 7c shows many digits of class '3', while Fig. 7d is less dominated by digit '3'.

There are 18 digits of '5' in Fig. 7a, while there are only 13 digits of '5' in Fig. 7b. However, Fig. 7a is actually dominated by '0', even though the base digit is a '5'. There are 54 digits of '0' in Fig. 7a, while there are only 25 digits of '0' in Fig. 7b. Similarly, there are 76 digits of '3' in Fig. 7c, while there are only 46 digits of '3' in Fig. 7d. We also observed that the image distance between Fig. 7a and Fig. 7b increased from 8.6 to 9.3, while the style distance decreased from 1.2 to 0.98. The image distance between Fig. 7c and Fig. 7d also increased from 8.5 to 9.5, while the style distance decreased from 1.2 to 1.0.

Similarly, there are 52 letters of 'S' in Fig. 8a, while there are only 6 letters of 'S' in Fig. 8b. Furthermore, there are 47 letters of 'P' in Fig. 8c, while there are only 17 letters of 'P' in Fig. 8d. The image distance between Fig. 8a and Fig. 8b increased from 9.1 to 10.5, while the style distance decreased from 1.3 to 0.91. Lastly, The image distance between Fig. 8c and Fig. 8d also increased from 8.5 to 9.8, while the style distance decreased from 1.3 to 0.99.

These results show that style memory successfully separates some of the class information from the data, while not being fully class-agnostic.

## 4.4 STYLE MEMORY INTERPOLATION

In this experiment, we attempted to reconstruct a continuum of images that illustrate a gradual transformation between two different styles of the same character class. For example, we encoded two different digits for each MNIST class, as shown in Fig. 9. We then generated a sequence of

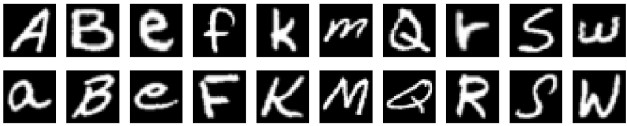

Figure 9: Two different styles of digits form the endpoints for the style interpolation experiment.

Figure 10: Two different styles of letters form the endpoints for the style interpolation experiment.

images that slowly evolve from one style to the other. We performed the interpolation by simply taking convex combinations of the two style memories, using

$$\hat{m}(\lambda) = \lambda m_1 + (1 - \lambda)m_2 \;, \tag{4}$$

where $m_1$ and $m_2$ denote the style memories. The interpolated style memory is denoted by $\hat{m}(\lambda)$, where $\lambda \in [0, 1]$ denotes the interpolation coefficient.

Figure 11 shows the interpolated digits and letters, illustrating that the generated images transform smoothly when the style memory is interpolated. The results of within-class interpolation suggest that style memory captures style information about how a digit was drawn. The figure also shows examples of attempted interpolations between incongruous letter forms (eg. 'A' to 'a', and 'r' to 'R'). Not surprisingly, the interpolated characters are nonsensical in those cases.

An obvious experiment is to try transferring the style memory of one digit onto another digit class. Although not shown here, we observed that the style memory of a digit can, in some cases, be transferred to some other classes. However, in general, the reconstructions did not look like characters.

## 5 CONCLUSIONS AND FUTURE WORK

Classification networks do not typically maintain enough information to reconstruct the input; they do not have to. Their goal is to map high-dimensional inputs to a small number of classes, typically using a lower-dimensional vector representation. In order for a classification network to be capable of generating samples, additional information needs to be maintained. In this paper, we proposed the addition of "style memory" to the top layer of a classification network. The top layer is trained using a multi-objective optimization, trying to simultaneously minimize classification error and reconstruction loss.

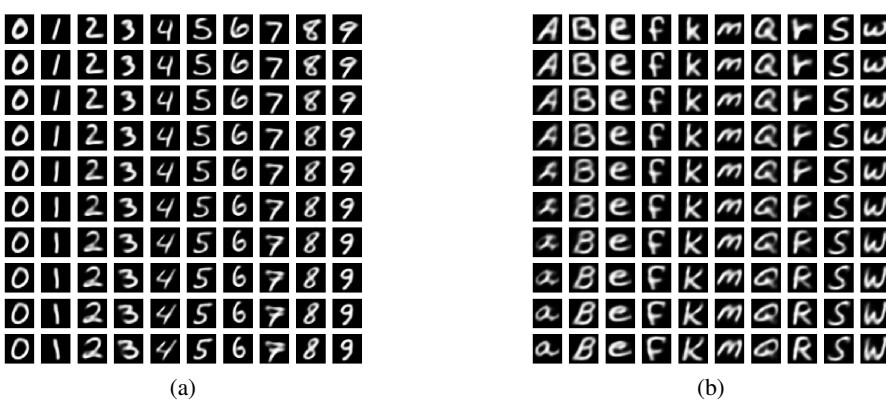

(a)                                    (b)

Figure 11: Image reconstruction with style memory interpolation between digits and letters shown in Fig. 9 and Fig. 10, where $\lambda$ was increasing from 0.1 to 1.0 with a step of 0.1 from top to bottom.

Our experiments suggest that the style memory encodes information that is largely disjoint from the classification vector. For example, proximity in image space yields digits that employ an overlapping set of pixels. However, proximity in style-memory space yielded a different set of digits.

For the style interpolation experiment, we generated images from a straight line in style-memory space. However, each position on this line generates a sample in image space – an image; it would be interesting to see what shape that 1-dimensional manifold takes in image space, and how it differs from straight-line interpolation in image space. However, the fact that we were able to interpolate digits and letters within the same class using novel style-memory activation patterns suggests that the style memory successfully encodes additional, abstract information about the encoded input.

To our knowledge, existing defence mechanisms to combat adversarial inputs do not involve the generative capacity of a network. Motivated by the results in Sec. 4.1, preliminary experiments that we have done suggest that treating perception as a two-way process, including both classification and reconstruction, is effective for guarding against being fooled by adversarial or ambiguous inputs. Continuing in this vein is left for future work.

Finally, we saw that the network has a property where the reconstruction generated was affected both by the classification neurons and style memory. Inspired by how human perception is influenced by expectation (Summerfield & De Lange, 2014), we believe that this work opens up opportunities to create a classifier network that takes advantage of its generative capability to detect misclassifications. Moreover, predictive estimator networks might be a natural implementation for such feed-back networks (Xu et al., 2017; Summerfield & De Lange, 2014; Orchard & Castricato, 2017). Perception and inference could be the result of running the network in feed-forward and feed-back directions simultaneously, like in the wake-sleep approach (Hinton et al., 1995). These experiments are ongoing.

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
