# OpenReview forum: "Style Memory: Making a Classifier Network Generative"
_ICLR.cc/2018/Conference — Reject_

### Official Review · AnonReviewer2 · 2017-11-22
**Lack of convincing motivation and results not particularly unimpressive**

**Rating:** 3
**Confidence:** 5

**Review:**

This paper proposes to train a classifier neural network not just to classifier, but also to reconstruct a representation of its input, in order to factorize the class information from the appearance (or "style" as used in this paper). This is done by first using unsupervised pretraining and then fine-tuning using a weighted combination of the regular multinomial NLL loss and a reconstruction loss at the last hidden layer. Experiments on MNIST are provided to analyse what this approach learns.

Unfortunately, I fail to see a significantly valuable contribution from this work. First, the paper could do a better job at motivating the problem being addressed. Why is it important to separate class from style? Should it allow better classification performance? If so, it's never measured in this work. If that's not the motivation, then what is it?

Second, all experiments were conducted on the MNIST dataset. In 2017, most would expect experiments on at least one other, more complex dataset, to trust any claims on a method.

Finally, the results are not particularly impressive. I don't find the reconstructions demonstrated particularly compelling (they are generally pretty different from the original input). Also, that the "style" representation contain less (and I'd say slightly less, in Figure 7 b and d, we still see a lot of same class nearest neighbors) is not exactly a surprising result. And the results of figure 9, showing poor reconstructions when changing the class representation essentially demonstrates that the method isn't able to factorize class and style successfully. The interpolation results of Figure 11 are also underwhelming, though possibly mostly because the reconstructions are in general not great. But most importantly, none of these results are measured in a quantitative way: they are all qualitative, and thus subjective.

For all these reasons, I'm afraid I must recommend this paper be rejected.

---

> ### Author Response · Authors · 2018-01-05
> **Response to AnonReviewer2**
>
> Thank you for your feedback. We apologize for not clearly stating the motivation behind this work. Our main motivation was to design a classifier network that also has the capacity to be generative. We believe that a generative network would be less susceptible to being fooled by adversarial inputs since it would not be able to reconstruct nonsensical input. But before showing that the network is less vulnerable to adversarial examples, we want to investigate the properties of such a network. We have added text to our paper to clarify that this is our ultimate goal.
>
> However, in creating a classifier/generative network, we wanted to investigate the relationship between the classification part of the encoding, and the “style memory” part of the encoding. Much of this paper is devoted to understanding this relationship.
>
> We have added experiments that we conducted on the Extended MNIST letter dataset which contains 145,600 samples, and where uppercase and lowercase letters are included in the same class (i.e. ‘A’ and ‘a’ are in the same class). This makes the dataset more challenging than MNIST. We also expanded our discussion of figure 7 (and figure 8, which was added). The discussion argues, in quantitative terms, that style memory contains a representation that augments, but is substantially different from, the character class. The figures also illustrate that the representation in style memory is very different from the original, image-space representation.
>
> We have modified our paper to address your comments, and feel that the paper is much improved from its original form.

---

### Official Review · AnonReviewer3 · 2017-11-27
**results are not convincing**

**Rating:** 3
**Confidence:** 5

**Review:**

The paper proposes training an autoencoder such that the middle layer representation consists of the class label of the input and a hidden vector representation called "style memory", which would presumably capture non-class information. The idea of learning representations that decompose into class-specific and class-agnostic parts, and more generally "style" and "content", is an interesting and long-standing problem. The results in the paper are mostly qualitative and only on MNIST. They do not show convincingly that the network managed to learn interesting class-specific and class-agnostic representations. It's not clear whether the examples shown in figures 7 to 11 are representative of the network's general behavior. The tSNE visualization in figure 6 seems to indicate that the style memory representation does not capture class information as well as the raw pixels, but doesn't indicate whether that representation is sensible.

The use of fully connected networks on images may affect the quality of the learned representations, and it may be necessary to use convolutional networks to get interesting results. It may also be interesting to consider class-specific representations that are more general than just the class label. For example, see "Learning a Nonlinear Embedding by Preserving Class Neighbourhood Structure" by Salakhutdinov and Hinton, 2007, which learns hidden vector representations for both class-specific and class-agnostic parts. (This paper should be cited.)

---

> ### Author Response · Authors · 2018-01-05
> **Response to AnonReviewer3**
>
> Thank you for your feedback. We apologize for not clearly stating the motivation behind this work. Our main motivation was to design a classifier network that also has the capacity to be generative. We believe that a generative network would be less susceptible to being fooled by adversarial inputs since it would not be able to reconstruct nonsensical input. But before showing that the network is less vulnerable to adversarial examples, we want to investigate the properties of such a network. We have added text to our paper to clarify that this is our ultimate goal.
>
> Although we have done preliminary experiments to show that the network is less vulnerable to adversarial examples, these results are not reported in this paper; we feel that further investigations and experimentations are warranted.
>
> Thank you for also pointing out the paper “Learning a Nonlinear Embedding by Preserving Class Neighbourhood Structure” by Salakhutdinov and Hinton (2007). We agree that the work is relevant, and have added a discussion of the paper to our “Related Work” section.
>
> We also changed our network to include convolutional layers, as you suggested. Lastly, we removed the tSNE visualization because we felt it did not service the main message of the paper.
>
> We have modified our paper to address your comments, and feel that the paper is much improved. We hope that you agree.

---

### Official Review · AnonReviewer1 · 2017-11-28
**The paper proposes augmenting classifier deep neural networks with 'style memory' features (along the lines of auto-encoders) and training the two at the same time.**

**Rating:** 4
**Confidence:** 3

**Review:**

The paper proposes combining classification-specific neural networks with auto-encoders. This is done in a straightforward manner by designating a few nodes in the output layer for classification and few for reconstruction. The training objective is then changed to minimize the sum of the classification loss (as measured by cross-entropy for instance) and the reconstruction error (as measured by ell-2 error as is done in training auto-encoders).

The authors minimize the loss function by greedy layer-wise training as is done in several prior works. The authors then perform other experiments on the learned representations in the output layer (those corresponding to classification + those corresponding to reconstruction). For example, the authors plot the nearest-neighbors for classification-features and for reconstruction-features and observe that the two are very different. The authors also observe that interpolating between two reconstruction-feature vectors (by convex combinations) seems to interpolate well between the two corresponding images.

While the experimental results are interesting they are not striking especially when viewed in the context of the tremendous amount of work on auto-encoders. Training the classification-features along with reconstruction-features does not seem to give any significantly new insights.

---

> ### Author Response · Authors · 2018-01-05
> **Response to AnonReviewer1**
>
> Thank you for your feedback. We apologize for not clearly stating the motivation behind this work. Our goal was to design a classifier network that also has the capacity to be generative. We believe that a generative network would be less susceptible to being fooled by adversarial inputs since it would not be able to reconstruct nonsensical input. But before showing that the network is less vulnerable to adversarial examples, we want to investigate the properties of such a network. However, results on adversarial inputs are not reported in this paper as we feel that further investigations and experiments are still needed.
>
> We have modified our paper to address your feedback. We feel our paper is much better after implementing those changes.

---

### Author Response · Authors · 2018-01-05
**Updated based on valuable reviewer comments**

We appreciate the constructive comments that the reviewers made on our paper, and have revised the manuscript accordingly. In particular, we have clarified the purpose of the research. This work is a necessary stepping-stone to our goal of investigating the possibility that generative networks are less susceptible to being fooled by ambiguous or adversarial inputs. The work outlined in this paper lays the foundation for how to create networks that simultaneously perform both classification and reconstruction. We have also included a more difficult dataset, EMNIST. We also altered the design of our network so that it now has two convolutional layers, and the resulting classification performance is much improved.

---

### Decision · Program_Chairs · 2018-01-29
**ICLR 2018 Conference Acceptance Decision**

**Decision:**

Reject

**Comment:**

 + Paper proposes simple joint deep autoencoder + classifier training where the hidden representation is split between (observed) class and (unobserved) style nodes.
 - Empirical evaluation is very limited, focusing on only qualitative evaluation of reconstructions and interpolations (on MNIST and EMNIST).
 - Unclear goal: if it is improving classifier robustness, then quantitative classifier robustness improvements should be experimentally demonstrated. If it is as a (conditional) generative model, then it should be compared to strong generative baselines (in the GVAE or GAN families). The paper currently has neither.